# Myoglobin–Pyruvate Interactions: Binding Thermodynamics, Structure–Function Relationships, and Impact on Oxygen Release Kinetics

**DOI:** 10.3390/ijms23158766

**Published:** 2022-08-06

**Authors:** Kiran Kumar Adepu, Dipendra Bhandari, Andriy Anishkin, Sean H. Adams, Sree V. Chintapalli

**Affiliations:** 1Arkansas Children’s Nutrition Center, Little Rock, AR 72202, USA; 2Department of Pediatrics, University of Arkansas for Medical Sciences, Little Rock, AR 72202, USA; 3Department of Biology, University of Maryland, College Park, MD 20742, USA; 4Department of Surgery, University of California Davis School of Medicine, Sacramento, CA 95817, USA; 5Center for Alimentary and Metabolic Science, University of California, Davis, CA 95817, USA

**Keywords:** pyruvate, myoglobin, oxygen release

## Abstract

Myoglobin (Mb), besides its roles as an oxygen (O_2_) carrier/storage protein and nitric oxide NO scavenger/producer, may participate in lipid trafficking and metabolite binding. Our recent findings have shown that O_2_ is released from oxy-Mb upon interaction with lactate (LAC, anerobic glycolysis end-product). Since pyruvate (PYR) is structurally similar and metabolically related to LAC, we investigated the effects of PYR (aerobic glycolysis end-product) on Mb using isothermal titration calorimetry, circular dichroism, and O_2_-kinetic studies to evaluate PYR affinity toward Mb and to compare the effects of PYR and LAC on O_2_ release kinetics of oxy-Mb. Similar to LAC, PYR interacts with both oxy- and deoxy-Mb with a 1:1 stoichiometry. Time-resolved circular dichroism spectra revealed that there are no major conformational changes in the secondary structures of oxy- or deoxy-Mb during interactions with PYR or LAC. However, we found contrasting results with respect to binding affinities and substrate preference, where PYR has higher affinity toward deoxy-Mb when compared with LAC (which prefers oxy-Mb). Furthermore, PYR interaction with oxy-Mb releases a significantly lower amount of O_2_ than LAC. Taken together, our findings support the hypothesis that glycolytic end-products play a distinctive role in the Mb-rich tissues by serving as novel regulators of O_2_ availability, and/or by impacting other activities related to oxy-/deoxy-Mb toggling in resting vs. exercised or metabolically activated conditions.

## 1. Introduction

Pyruvate (PYR) is a product of glycolysis and a primary substrate for oxidative metabolism to support tissue energy needs [1]. In humans, the physiological concentration of PYR ranges from 0.1 to 0.7 µmol/g in skeletal muscle and 30 to 260 µM in blood [2,3,4]. Mitochondria play a key role in energy metabolism, importing PYR from the breakdown of glucose through the mitochondrial pyruvate carrier (MPC). During hypoxia, when O_2_ levels are low and anerobic glycolysis is prevalent, PYR is reduced to lactate (LAC) by lactate dehydrogenase (LDH) in the cytosol [5,6,7]. In accordance, skeletal muscle (especially type II) is described as a robust LAC-producing tissue, whereas heart is considered as a PYR-oxidizing organ.

For several anabolic and catabolic pathways, PYR metabolism serves as a branching point for oxidative metabolism, maintenance of tricarboxylic acid cycle (TCA cycle), re-synthesis of glucose (gluconeogenesis), lipid synthesis (de novo lipogenesis), and cholesterol synthesis. At the whole-body level, at least six different pathways influence PYR pools and utilization, e.g., (i) actions of lactate dehydrogenase that regenerates PYR from LAC for use as fuel or biosynthetic substrate, (ii) generation from malate through malic enzyme, (iii) catabolism of 3-carbon amino acids (iv), pyruvate dehydrogenase complex that generates acetyl-CoA, (v) pyruvate carboxylase reaction that supplies oxaloacetate, and (vi) acetate biosynthetic pathway that converts PYR directly to acetate [8,9,10]. The majority of cytosolic PYR is imported into the mitochondria adenosine triphosphate (ATP) production via oxidative phosphorylation with multiple biosynthetic pathways intersecting the TCA cycle [5].

Previous studies have highlighted that myoglobin (Mb) is not limited to a role in oxygen (O_2_) storage and O_2_ trafficking, but other functions such as nitric oxide (NO) scavenging/production and lipid peroxidation are important [11,12,13,14,15,16,17,18,19,20,21,22]. In addition, we and others have shown that Mb interacts with fatty acids [23,24,25,26,27,28] and long-chain acylcarnitine derivatives [29,30], specifically to oxy-Mb but not deoxy-Mb. Other studies have proposed that LAC binds to Oxy-Mb (in high LAC conditions e.g., from 1 mM to 10 mM) and reduces the O_2_ affinity of Mb to release O_2_ inside the cell [31]. Recently, we confirmed that cellular LAC concentrations influence O_2_ release upon binding to oxy-Mb [32]. Depending on the pH conditions, LAC interacts with both oxy- and deoxy-Mb with different affinities [33]. Specifically, at neutral pH 7.0, LAC shows interaction only with oxy-Mb but not to deoxy-Mb. However, under acidic conditions, LAC interacts with both oxy- and deoxy-Mb structures coincident with release of O_2_ from the oxy-Mb. Considering both PYR and LAC are end-products of glycolysis, and given that PYR and LAC share structural similarity, it is reasonable to consider that PYR may also bind to both oxy and deoxy-Mb structures and influence O_2_ release from Mb. To address these questions, we performed isothermal titration calorimetry (ITC), O_2_ kinetic evaluation, and circular dichroism (CD) spectroscopic studies. The current study will help in understanding the effect of glycolytic end-products on Mb-O_2_ dynamics and may unravel novel physiological events linking oxidative and non-oxidative glucose metabolism to Mb activities and oxygenation states.

## 2. Results and Discussion

Isothermal titration calorimetry (ITC) binding studies revealed that PYR binds to Mb at a 1:1 molar ratio stoichiometry. A similar observation was also observed for binding of LAC with Mb [32]. Figure 1 displays the binding profile of Mb-PYR at different pH conditions. In Figure 1, sub-figures include raw binding data (upper panel) of measured potential difference (DP in µcal/s) and processed binding data (lower panel) of change in enthalpy (ΔH in kcal/mol). Unlike LAC, PYR binds to both oxy-Mb and deoxy-Mb with high affinity to the latter at neutral and acidic pH (Table 1). Earlier studies on LAC interaction with Mb showed that LAC prefers to bind oxy-Mb at neutral pH, and highest affinity (*K_d_* = 1.9 ± 0.2 µM) was observed at pH 6.0 followed by pH 6.4 (*K_d_* = 6.9 ± 1.1 µM) [32]. Whereas in PYR binding, highest affinity (*K_d_* = 0.71 ± 0.11 µM) was observed with deoxy-Mb at pH 7.0. Based on the *K_d_* values, PYR binding to the Mb protein was in the order of deoxy-Mb (pH 7.0) > deoxy-Mb (pH 6.4) > oxy-Mb (pH 6.4) > deoxy-Mb (pH 6.0) > oxy-Mb (pH 7.0) > oxy-Mb (pH 6.0), respectively.

At pH 7.0, an upward slope pattern is observed when oxy-Mb interacts with PYR (Figure 1a). Moreover, since enthalpic (∆H) values are negative (Table 1), the binding is favored by an exothermic reaction i.e., heat is released during the interaction of Mb and PYR. Similar observations were also found with deoxy-Mb interaction with PYR at pH 7.0 and pH 6.4 (lower panels of Figure 1d,e). To support the measurements, signature plot analysis of these data also support that the interactions between Mb and PYR were driven by change in enthalpy (∆H) and hydrophilic interactions stabilize the Mb+PYR complex (Figure 2). In contrast, at acidic pH, the oxy-Mb+PYR complex was favored by hydrophobic interactions with large positive enthalpic (∆H) values, and the complex was driven by change in entropy (∆H) (Table 1). In acidic pH conditions, the protonation state of Mb protein is changed [33] and this could be one of the possible reason for the observed difference in interaction of Mb and PYR at neutral and acidic pH. ITC studies also revealed the binding affinity (*K_a_*, inverse of *K_d_*) of PYR interaction with oxy- and deoxy-Mb in different pH conditions. Specifically, with a drop in pH, the *K_a_* of PYR for deoxy-Mb was decreased by ~7-fold (at pH 6.4) and ~60-fold (at pH 6.0) relative to pH 7.0 value (Table 1). 

O_2_ release kinetic studies with varying concentrations of PYR at neutral and acidic pH conditions showed that the addition of increasing concentrations of PYR to oxy-Mb resulted in an increase in the release of O_2_ at neutral pH (Figure 3a**)**. However, at acidic pH, the addition of PYR to oxy-Mb did not release O_2_ compared to oxy-Mb alone (Appendix A). Irrespective of pH, with oxy-Mb solution alone (no PYR present, i.e., pre-PYR), little to no change in the O_2_ levels was observed (inset of Figure 3b). This confirms that O_2_ release from oxy-Mb at pH 7.0 is due to a specific interaction of PYR with oxy-Mb. In contrast to PYR, our recent studies showed that the rate of release of O_2_ coincident with LAC presence was significant at acidic pH levels [32]. This is likely due to differential affinities for the oxy-Mb/metabolite interactions at lower pH: oxy-Mb’s affinity toward LAC (*K_d_* value of 1.9 µM [*K_a_* value of 526 nM] at pH 6.0) is far higher than that for PYR (*K_d_* value of 101 µM at pH 6.0). Additionally, PYR has high affinity (*K_d_* value of 0.71 µM [*K_a_* value of 1408 nM] at pH 7.0) towards deoxy-Mb. A maximum rate of O_2_ release (70 nmol/min/g protein) from oxy-Mb was observed with 5.0 mM PYR in pH 7.0 buffer (Figure 3b). Intriguingly, a lower concentration of LAC (2.5 mM) was required to release a similar rate of O_2_ from oxy-Mb at pH 6.4 [32]. The complex nature of pH-dependent changes in O_2_ release from oxy-Mb with PYR or LAC might be due to minor conformational changes in the Mb structure with respect to pH. These open questions need further comprehensive investigation. 

Computational analysis revealed that Mb protein acid dissociation constant (p*K_a_*) was increased with decreasing pH, i.e., p*K_a_* of ~2.6, ~6.3, and ~8.9 at pH 7.0, pH 6.4, and pH 6.0, respectively. Apart from the net charge, another compelling reason might be the position of the binding site region of PYR during the interaction with Mb in different pH conditions (Figure 4). Molecular docking studies revealed that PYR may bind to deoxy-Mb at different locations depending on changes in pH. However, with oxy-Mb, irrespective of pH, no change in the region of PYR binding was observed. PYR binds near the loop region between helices A and G via hydrogen bonding interactions with the residues D121 and D126 away from the heme center (Figure 4a–c). In contrast, at pH 6.0, PYR binds deoxy-Mb at residue R31 of helix B (Figure 4f), whereas at pH 7.0 and pH 6.4, PYR interacts with residues W14, E18, and K77 of helices A and E of deoxy-Mb (Figure 4d,e). 

To the best of our knowledge, this is the first report on Mb–PYR interaction. Interestingly, docking studies resulted in entirely different binding regions for PYR and LAC. The presence of a ketone group instead of hydroxyl at the second carbon of PYR makes this position significantly less hydrophilic compared to LAC and decreases the ability to participate in H-bonding, which might explain the changes in the preferred binding regions and Mb oxygenation states. In our earlier docking study, at pH 7.0, LAC was found to bind to oxy-Mb near the O_2_ binding site (proximal His residue) of the heme center [32]. Docking studies have also shown that LAC interacts with the residues K45, D60, and K63 of oxy-Mb (Appendix A). Whereas, in decreasing pH conditions to pH 6.4, LAC binds in a different region away from the heme center and interacts with residues K41 and K97 of oxy-Mb (Appendix A). However, at pH 6.4, although LAC was found to bind near the heme center of oxy-Mb, LAC. At pH 6.0, LAC interacts with residues K56 and E59 of oxy-Mb away from heme binding center (Appendix A). With deoxy-Mb, LAC was found to bind near the proximal His side of the heme center interacting with the residues H96 and S92, except at pH 7.0, where no binding was observed (Appendix A). For comprehensive structural details of LAC interaction with Mb, readers are suggested to refer to our recent published report [32]. More computational and simulation studies are needed to reveal the structure–function relationship properties of the Mb+PYR complexes at different pH conditions. However, the present ITC binding results and molecular docking predictions strengthen our hypotheses presented in Figure 5 (discussed in later sections) related to different affinities of oxy-Mb and deoxy-Mb toward PYR in varied pH conditions. In parallel, ITC binding experiments using a non-globin protein, LYZ, showed no detectable binding with PYR in all the tested pH conditions (Appendix A). This supports the idea that there is a degree of specificity to the PYR binding to Mb. 

Further, time-resolved circular dichroism spectra revealed that PYR or LAC interaction with oxy- or deoxy-Mb did not lead to major conformational changes in the secondary structure (α-helix, β-sheet, turns) of the protein at varying pH and monocarboxylate concentrations (Table 2, Appendix A). However, comparing PYR and LAC interaction with Mb, the α-helical content of oxy-Mb was marginally decreased with LAC at pH 7.0. Here, the term ‘decrease’ means the protein conformational change observed in either α, β or disordered to one another. Although similar observations were recorded with deoxy-Mb at pH 6.4, no significant differences between PYR and LAC in α-helical region of the oxy- and deoxy-Mb protein were observed at pH 6.4 and 6.0. However, significant differences were observed between PYR and LAC in α-helical region of the oxy-Mb at pH 7.0. CD spectra clearly revealed that significant changes in Mb protein were only at the disordered regions (“Others” in Table 2). This is expected, because PYR (Figure 4) and LAC (Figures 2 and 3 in [32]) bind in the loop region of oxy-Mb.

Recently, we showed that LAC does not bind to deoxy-Mb at pH 7.0 and binds only to oxy-Mb at neutral and acidic pH and rapidly releases O_2_ in the latter [32]. Intriguingly, irrespective of pH, LAC prefers to bind oxy-Mb, while PYR prefers deoxy-Mb. It is possible that the shifts in these metabolites regulate O_2_ availability and trafficking through Mb. Alternatively, it has been speculated that Mb also serves as an as an O_2_ “sensor” that regulates oxidative phosphorylation via regulating NO pools [34], and we have hypothesized that O_2_-sensitive toggling between oxy- and deoxy-Mb modifies cellular signaling pathways and gene expression [35]. Might interaction with metabolites such as fatty acids, acylcarnitines, LAC, or PYR modify these activities? Further comprehensive investigations are needed to address such crucial questions. In exercising muscle or tissue actively generating cellular LAC there is a concomitant reduction in the intracellular pH from pH 6.8–7.2 to pH 5.0–6.5. Moreover, it was reported that compared to resting muscle, the LAC/to PYR ratio reaches ≥ 80 during intense exercise [36,37]. Considering the above findings, we support the working model that LAC preferential binding to oxy-Mb releases O_2_ and converts Mb into deoxy-Mb; thereafter, PYR binding to deoxy-Mb may limit LAC binding, which would tend to promote O_2_ binding and oxy-Mb conversion. A schematic representation of this hypothesis on the biochemical events in resting verses working muscles is shown in Figure 5.

## 3. Materials and Methods

### 3.1. Materials

Horse heart muscle myoglobin (Mb), chicken egg white lysozyme (LYZ), sodium pyruvate (PYR), sodium lactate (LAC), and sodium dithionite were purchased from Sigma Aldrich, St. Louis, MO, USA. All other chemicals used in the experiments are of analytical grade and were also procured from Sigma Aldrich, St. Louis, MO, USA.

### 3.2. Preparation of Mb

Preparations enriched in oxygenated and deoxygenated forms of Mb were prepared as described by our earlier publications related to Mb interaction with LAC, fatty acids, and acylcarnitines [30,32]. In brief, 500 µM of Mb was dissolved in 50 mM sodium phosphate buffer of desired test pH. To promote the conversion of ferric (Fe^3+^) to ferrous (Fe^2+^) iron, 3 mM sodium dithionite was added to the protein solution with gentle mixing. Thereafter, the solution was subjected to a desalting column to remove the reducing agent, to remove the interference particularly in O_2_ release kinetics and UV-Vis spectroscopy absorbance peaks. Thereafter, it was purged with either O_2_ or N_2_ gas continuously for 10 min into the protein solution enriched in oxy- and deoxy-Mb, respectively. The formations of oxy- and deoxy-Mb were confirmed based on their characteristic peaks using UV-visible spectroscopy.

### 3.3. Ligand Binding Studies

Protein–ligand binding experiments were performed using isothermal titration calorimetry (ITC) (Microcal PEAQ-ITC, Malvern Instruments Ltd, Malvern, UK). ITC studies were performed in the presence of sodium dithionite, as it did not show any effect on binding properties. Before starting the experiment, both the ligand solution and the protein solution were purged with either O_2_ or N_2_ for 10 min. Both the protein and PYR solutions were thermally equilibrated to 25 °C prior the start of the titration. PYR was loaded in the reaction cell at an initial concentration of 50 µM and titrated against 500 µM of either oxy- or deoxy-Mb solution, maintaining 1:10 ratio between ligand and protein. A total of 19 injections (2 µL each) from the syringe were used to generate the ITC curves within each experiment. During the experimental run, the samples were mixed thoroughly at constant stirring of 750 rpm. Between each injection, a 150 s gap was maintained to achieve a stable baseline. Data obtained from the ITC experiments were best fit to a one-set of sites binding model provided by the Microcal PEAQ-ITC software (version 1.40). Heats of dilution and heats due to potential products formed during the time of the ITC experiments were corrected by performing appropriate blank titrations, consisting of (a) either oxy- or deoxy-Mb into the test buffer solution, (b) test buffer into the PYR, and (c) buffer-buffer solution. Lysozyme was used as a negative control protein in the protein–ligand binding experiments. Comparing with our earlier results obtained from Mb–LAC interaction [32], ITC experiments were also performed at three different pH conditions (7.0, 6.4 and 6.0), mimicking intracellular physiological and acidic pH states that would be typically observed in skeletal muscle cells in “rested” and “active” conditions. All the binding experiments were performed three times (*n* = 3) and data obtained from statistical analysis are presented herein.

The change in entropy (ΔS) was calculated using the equation:(1)$$\Delta S\;=\frac{\Delta H\;-\;\Delta G}{T}$$
where ΔG represents the change in Gibbs free energy, ΔH is the change in enthalpy, and T is the absolute temperature.

*c* values were calculated using the equation:*c* = *nK_a_*[M]_t_(2)
where *n* is the number of binding sites per receptor (macromolecule), [M]_t_ is macromolecule concentration, and *K_a_* is the association constant. *c*-value determines the accuracy of curve fitting to obtain *K_d_* and binding stoichiometry [38]. 

### 3.4. Oxygen Release Kinetic Studies

The O_2_ concentrations (release and binding) during the ligand interactions with Mb in the solution were measured using Oxytherm+ liquid-phase oxygen electrode system (Hansatech Instruments, Norfolk, UK). The Oxytherm+ respirometer is an advanced instrument routinely used for respiration studies. 

The measurement of dissolved O_2_ is calculated at the given temperature and atmospheric pressure according to the following equation [39]:Cs = 14.16 − (0.394 × T) + (0.007714 × T^2^) − (0.0000646 × T^2^)(3)
where Cs is the saturated O_2_ concentration in ppm and T is the temperature in °C. One part per million is equivalent to 1 µg/mL or (1 µg/32 g/mol) = 0.03125 µmol/mL or 31.25 nmol/mL.

The optimum concentration of oxy-Mb was found to be 25 µM, and PYR concentrations were varied from 500 µM to 5 mM. Similarly, deoxy-Mb protein alone (i.e., without PYR) was also tested. All the experiments were carried out at a constant temperature of 25 °C using a Peltier thermostat. The solutions were mixed at 50 rpm using a small magnetic stirring bar placed inside the sample container. Samples were injected into the buffer solution (50 mM sodium phosphate) at varying pH levels (pH 6.0–7.0) using a Hamilton glass syringe (1 cc) after achieving equilibrium. Ligand was added to the Mb protein. Appropriate buffer controls voiding either Mb or PYR were used to nullify any artefacts. O_2_ kinetic experiments were also performed. All the kinetic experiments were performed three times (*n = 3*) and data statistical analysis are presented herein.

### 3.5. Circular Dichroism Spectroscopic Studies

The change in spectral characteristics of equine Mb alone and PYR-bound Mb were recorded using CD spectropolarimeter (J-1500 model, Jasco Instruments, Easton, MD, USA) under constant nitrogen atmosphere (10 mL/min). All the sample preparations were similar to the ITC experiments. After degassing Mb and PYR samples either with O_2_ or N_2_ gas in sealed vials, appropriate volumes of the samples were taken using a Hamilton glass syringe (1 cc) and mixed in a quartz cuvette (1 mm) and sealed immediately without any delay; CD spectra were recorded at 37 °C between wavelengths of 190–260 nm. Since our earlier studies showed that LAC binds to Mb, CD spectral experiments were also conducted using LAC as a comparative monocarboxylate ion to detect any structural changes of Mb protein. Appropriate buffer controls void of either Mb or PYR were used, and these blank values were subtracted from values derived from treatments containing Mb and PYR. All the spectral experiments were performed three times (*n* = 3) and averaged to obtain the final spectrum. The *α*-helix, *β*-sheet, and random coil contents were determined by using the integrated Jasco application software (quantitative multi-variate analysis), PLS algorithm provided by the manufacturer.

### 3.6. Molecular Docking

Autodock 4.2 [40] was used to dock PYR to the heme-binding pocket of deoxy-Mb (PDB: 2V1K). We used the relaxed model of the oxy-Mb structure derived from horse deoxy-Mb (due to the unavailability of the oxy-Mb crystal structure), which is used in our previous studies [29]. A three-dimensional structure of PYR was obtained from PubChem database (https://pubchem.ncbi.nlm.nih.gov/compound/Pyruvate/, accessed on 11 June 2022). The iron ion parameters in the heme group were obtained from Autodoc 4.2 software. The iron ion was selected directly from the Set Map Types within the Grid tab to add iron with default parameters for docking. The protonation state of each titratable residue in Mb at different pH values was set based on p*K_a_* estimations by PROPKA [33]. AutoDock Tool was then used to prepare the protein–ligand system by assigning polar hydrogen atoms and Kollman’s partial charges with solvation parameters to the protein. A grid box of search space (70 Å × 70 Å × 70 Å) enclosing the heme group and residues within 5 Å from the heme center with a grid spacing of 0.375 Å was assigned. A Lamarckian genetic algorithm (LGA) was applied with a population size of 300 and 25 million maximum energy evaluations for 150 independent runs. The best docking structure is selected based on the lowest binding energy within the largest cluster of the docking results.

### 3.7. Statistical Analysis

Statistical analysis was performed using Microcal Origin software via an iterative algorithm for all ITC binding experiments. Similarly, nonlinear regression analysis of the average data points was calculated for each condition. One-way ANOVA was performed for oxy-Mb and deoxy-Mb conditions separately to determine the statistically significant differences between the data obtained from the binding studies in different pH conditions with a level of confidence of 95%. For oxygen kinetic analysis, a one-way ANOVA and the Tukey–Kramer post hoc test were performed to determine the statistical significance. All the experimental results data are presented as means ± standard error (SE).

## 4. Conclusions

In the present study, we showed that PYR interacts with both oxy- and deoxy-Mb at neutral and acidic pH conditions. Binding and spectroscopic studies revealed that PYR has a higher affinity toward deoxy-Mb compared to oxy-Mb, a pattern that contrasts with LAC, which has a higher affinity toward oxy-Mb. Similarly, docking models revealed that PYR interacts with deoxy-Mb at varying sites in a pH-dependent manner (pH 6.4 verses pH 6.0), whereas with oxy-Mb, the PYR is confined to one binding site irrespective of the changing pH. It is reasonable to speculate that the cellular concentrations of these metabolites, coupled to changes in pH, play a major role in O_2_ affinity to Mb and thus the natural toggling between oxy- and deoxy-Mb. Future studies involving quantum/molecular (QM/MM) mechanics are warranted to study the mechanism of O_2_ release upon binding of LAC and PYR to oxy-Mb, and whether or not coincident binding by long-chain fatty acids, long-chain acylcarnitines or other metabolites modify these outcomes. In the future, site-directed mutagenesis could be considered to determine the importance of specific amino acids residues involved with PYR and LAC binding, identified through simulation studies.

Overall, the results presented here and in our recent publication [32] indicate that glycolytic end products, PYR and LAC, interact with Mb with distinct pH-dependent binding characteristics. The novel observation that PYR prefers deoxy-Mb binding and LAC favors oxy-Mb binding suggests an important role of these metabolites in the physiological function of Mb in altering tissue oxygenation, which can have important implications for cellular bioenergetics during resting versus intense exercise (hypoxic) conditions. Alternatively, through oxy-Mb/deoxy-Mb, toggling the Mb protein has been hypothesized to be an O_2_-sensitive regulator of cytochrome c oxidase [41], and/or oxy-/deoxy-Mb may regulate signaling pathways that regulate outcomes such as gene expression [35]. Whether or not binding to one or more metabolites such as LAC, PYR, fatty acids, or acylcarnitines influences these non-canonical roles for Mb remains to be tested. 

## Figures and Tables

**Figure 1 ijms-23-08766-f001:**
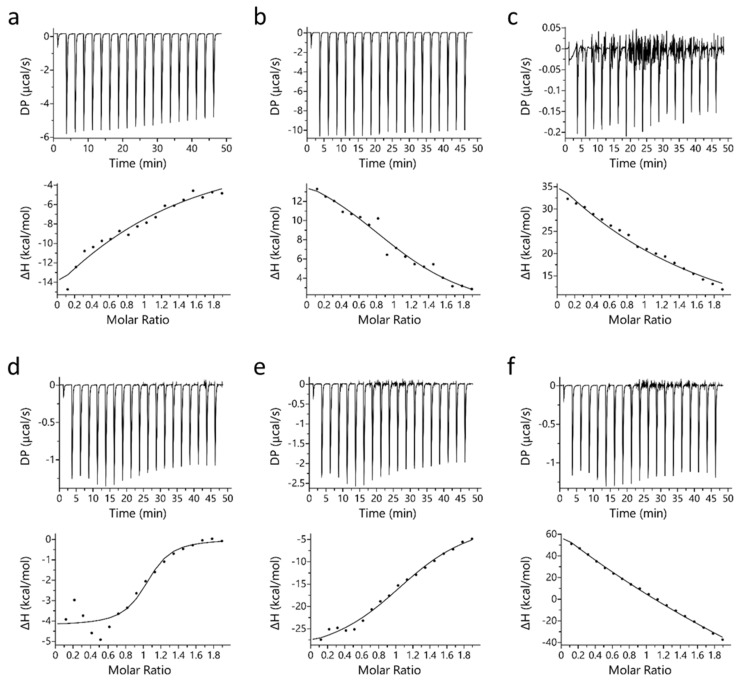
Representative ITC plots of binding of PYR with equine Mb. Top row (**a**–**c**) displays PYR interaction with oxy-Mb at (**a**) pH 7.0, (**b**) pH 6.4, and (**c**) pH 6.0. Bottom row (**d**–**f**) displays PYR interaction with deoxy-Mb at (**d**) pH 7.0 (**e**) pH 6.4, and (**f**) pH 6.0. In each sub-figure, raw data (upper panels) and integrated data (lower panels) represent titration of reactants with time (min) or molar ratios on the *x*-axis and the energy released or absorbed per injection on the *y*-axis. The solid lines in the bottom panels represent the best-fit of experimental data using ‘one-set of sites’ model provided by the manufacturer’s software (Microcal PEAQ-ITC software). The lower graphs clearly differentiate that for oxy- and deoxy-Mb pH 7.0, and deoxy-Mb at pH 6.4, Mb-PYR binding was predominantly exothermic (upward slope) driven by hydrophobic interactions (details are given in Section 2). In contrast, at acidic pH (pH 6.4 and pH 6.0), the Mb–LAC binding was endothermic (downward slopes) and mostly favored by hydrophilic interactions. All the ITC experiments were repeated 3 times (*n* = 3) to obtain the thermodynamic properties. Statistical analysis was performed using one-way ANOVA. Shown here is one representative dataset from a single experiment per condition. In each sub-figure, heat flow (DP) and change in enthalpy (ΔH) are shown here.

**Figure 2 ijms-23-08766-f002:**
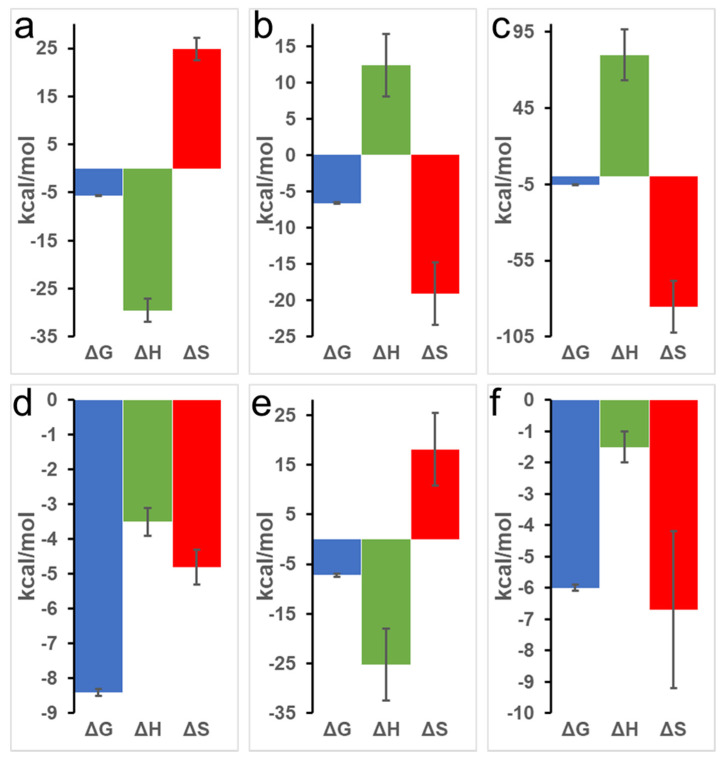
Representative signature plots of binding of PYR with equine Mb. Top row (**a**–**c**) displays PYR interaction with oxy-Mb at (**a**) pH 7.0, (**b**) pH 6.4, and **(c**) pH 6.0. Bottom row (**d**–**f**) displays PYR interaction with deoxy-Mb at (**d**) pH 7.0, (**e**) pH 6.4, and (**f**) pH 6.0. All the ITC experiments were repeated 3 times (*n* = 3) to obtain the thermodynamic properties. Statistical analysis was performed using one-way ANOVA.

**Figure 3 ijms-23-08766-f003:**
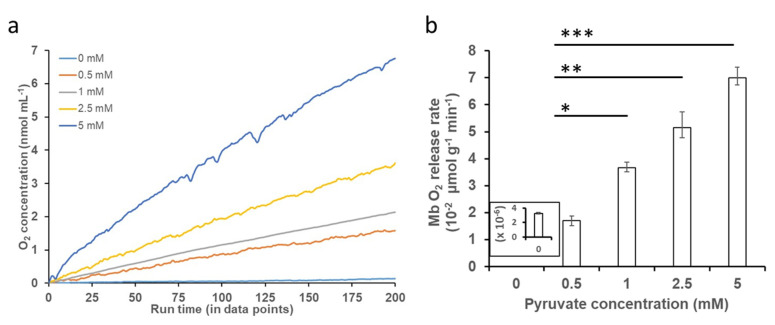
Effect of PYR binding to oxy-Mb and O_2_ release. O_2_ release kinetics were studied using an Oxytherm+ respirometer. All experiments were performed in 50 mM sodium phosphate buffer (pH 7.0, pH 6.4 and pH 6.0) containing 150 μM of oxy-Mb-enriched equine Mb preparations and varying concentrations of PYR (0.5 mM to 5 mM), using oxy-Mb alone as a zero-PYR control. (**a**) Representative graph showing O_2_ release from oxy-Mb after addition of varying concentrations of PYR at pH 7.0. No release of O_2_ from oxy-Mb in acidic pH (pH 6.0–pH 6.4) was detected (Appendix A). (**b**) Rate of release of O_2_ from Mb against PYR concentrations at pH 7.0, calculated from the linear portion of the graphs after addition of PYR to oxy-Mb. All O_2_ experiments were repeated 3 times (*n* = 3). Statistical analysis was performed using one-way ANOVA. * *p* < 0.05, ** *p* < 0.01, *** *p* < 0.001.

**Figure 4 ijms-23-08766-f004:**
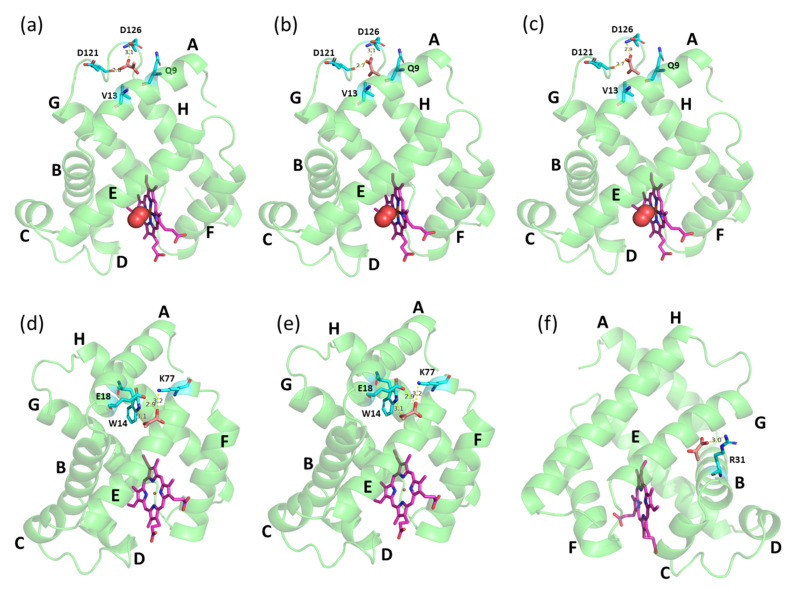
Docking structures of PYR binding with equine skeletal muscle Mb. Top row displays oxy-Mb at (**a**) pH 7.0, (**b**) pH 6.4 and (**c**) pH 6.0, and bottom row displays deoxy-Mb at (**d**) pH 7.0, (**e**) pH 6.4 and (**f**) pH 6.0, respectively. PYR (brown), heme center (pink), and the amino acid residues (cyan) interacting with PYR are displayed as sticks. Mb protein (green) is displayed as ribbon structure an oxygen (red) in spheres. Possible hydrogen bond interactions between side chains of residues and PYR are displayed as dashed yellow lines with bond length in angstroms.

**Figure 5 ijms-23-08766-f005:**
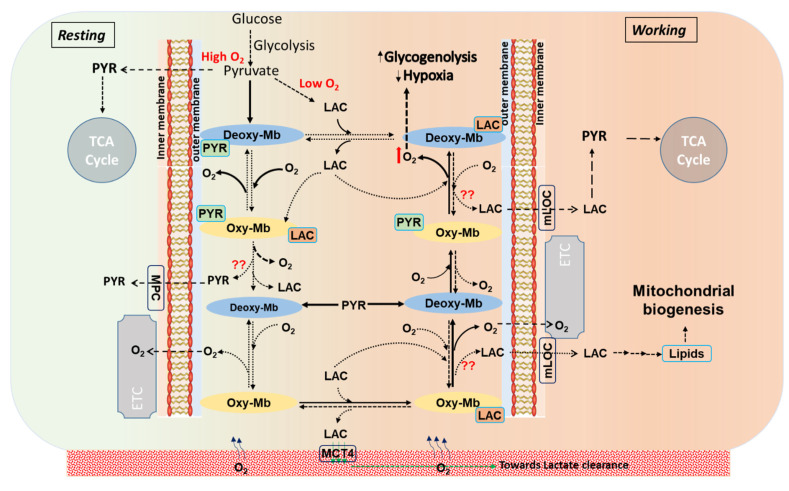
A schematic representation displays the stoichiometric turnover events on conversion of oxy-myoglobin (oxy-Mb) to deoxy-myoglobin (deoxy-Mb) and vice versa in metabolite-bound Mb states in a normal aerobic cell (resting) at constant supply of diffused oxygen (O_2_), or in working muscle characterized by reduced pO_2_ and lower pH. In resting conditions (displayed in light green color background, leftward), aerobic glycolysis generates PYR that avidly binds to deoxy-Mb (shown in solids arrows). Similarly, LAC (end-product of anaerobic glycolysis), increasingly generated as workload increases and pO_2_ drops (displayed in light red color background, rightward), binds avidly to oxy-Mb, but does not bind to deoxy-Mb [32]. However, in working muscle cells and at elevated lactic acid (LAC) levels (>1.0 mM), although PYR binds to both oxy-Mb and deoxy-Mb with low binding affinities, LAC strongly binds to Mb in acidic conditions and releases O_2_ from oxy-Mb [32]. In all these proposed events, release of Mb-bound PYR/LAC is not clearly known. Solid arrows (
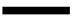
) display high affinity and rapid reaction rates and dashed arrows (
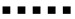
) display lower affinity and slower reaction rates. oxy-Mb: oxygenated-Mb; deoxy-Mb: deoxygenated-Mb; PYR: pyruvate; LAC: lactate; TCA cycle: tricarboxylic acid cycle; ETC: electron transport chain; MPC: mitochondrial pyruvate complex; mLOC: mitochondrial lactate oxidation complex; MCT4: monocarboxylate transporter 4.

**Table 1 ijms-23-08766-t001:** Binding thermodynamics of pyruvate (PYR) titrated against equine oxy-Mb and deoxy-Mb in ITC experiments at different pH conditions. Dissociation constant (*K_d_*), association constant (*K_a_*), Gibbs free energy (ΔG), change in enthalpy (ΔH), change in entropy (ΔS), and the accuracy of curve fitting to obtain *K_d_* and binding stoichiometry (*c*-value) are shown in the table. All the ITC experiments were repeated 3 times (*n* = 3) to obtain the thermodynamic properties. Statistical analysis was performed using one-way ANOVA for each parameter, within oxy-Mb or deoxy-Mb at each pH condition, and the values presented as means ± SEM. Numerical and alphabetical data in the superscript represent statistical significance (*p*-value < 0.05) among the test pH conditions for oxy- and deoxy-Mb, respectively.

Thermal Properties	pH 7.0		pH 6.4		pH 6.0	
	Oxy-Mb	Deoxy-Mb	Oxy-Mb	Deoxy-Mb	Oxy-Mb	Deoxy-Mb
*K_d_* (µM)	77.1 ± 1.7 ^1^	0.71 ± 0.11 ^a^	14.2 ± 1.1 ^2^	5.7 ± 1.5 ^b^	100.8 ± 4.3 ^3^	42.4 ± 5.9 ^c^
*K_a_* (nM)	12.9 ± 0.3 ^1^	1474 ± 211 ^a^	71.02 ± 5.0 ^2^	218 ± 80 ^b^	9.95 ± 0.4 ^3^	24.4 ± 3.2 ^c^
*c* value	6.5 ± 0.1 ^1^	737 ± 105 ^a^	35.5 ± 2.5 ^2^	109 ± 40 ^b^	4.9 ± 0.2 ^3^	12.2 ± 1.6 ^c^
ΔG (kcal mol^−1^)	−5.6 ± 0.1 ^1^	−8.4 ± 0.1 ^a^	−6.6 ± 0.1 ^2^	−7.2 ± 0.3 ^b^	−5.2 ± 0.2 ^1^	−6.0 ± 0.1 ^b^
ΔH (kcal mol^−1^)	−29.5 ± 2.4 ^1^	−3.5 ± 0.4 ^a^	12.4 ± 4.3 ^2^	−25.2 ± 7.2 ^b^	79.9 ± 16.7 ^3^	−1.5 ± 0.5 ^a^
ΔS (cal mol^−1^ K^−1^)	24.8 ± 2.3 ^1^	−4.8 ± 0.5 ^a^	−19.1 ± 4.3 ^2^	18.1 ± 7.3 ^b^	−85.5 ± 17 ^3^	−6.7 ± 2.4 ^a^
No. of binding sites	0.99 ± 0.03 ^1^	0.98 ± 0.02 ^a^	0.81 ± 0.05 ^1^	0.88 ± 0.03 ^a^	0.92 ± 0.07 ^1^	1.03 ± 0.04 ^a^

**Table 2 ijms-23-08766-t002:** Changes in secondary structure conformations (%) of Mb with varying concentrations of metabolite, PYR, and LAC.

Protein State	Metabolite	Metabolite Concentration	pH 7.0				pH 6.4				pH 6.0			
			*α*-Helix	*β*-Sheet	Turn	Others	*α*-Helix	*β*-Sheet	Turn	Others	*α*-Helix	*β*-Sheet	Turn	Others
Oxygenated-Mb		Control (0 mM)	78.4	0.0	4.2	17.4	77.9	0.0	15.4	6.7	77.7	0.0	7.8	14.5
	PYR	1 mM	84.1	0.0	6.0	9.9	74.2	0.0	6.2	19.6	64.2	3.0	9.0	23.8
		2 mM	75.7	0.0	5.1	19.2	70.2	0.0	13.5	16.3	70.0	0.2	8.5	21.3
		3 mM	80.7	0.0	6.8	12.5	73.3	0.0	10.1	16.6	75.7	0.0	9.3	15.0
		4 mM	79.4	0.0	7.3	13.3	78.6	0.0	3.5	17.9	67.9	7.6	11.5	13.0
		5 mM	77.9	0.0	7.0	15.1	77.2	0.0	9.6	13.2	75.7	0.0	7.1	17.2
		6 mM	80.5	0.0	11.4	8.1	79.9	0.0	10.8	9.3	78.4	0.0	11.9	9.7
		7 mM	80.2	0.0	6.6	13.2	76.7	0.0	10.0	13.3	73.1	0.0	11.6	15.3
		8 mM	78.9	0.0	5.5	15.6	76.1	0.0	4.6	19.3	72.5	0.0	12.1	15.4
	LAC	1 mM	68.5	0.0	13.2	18.3	75.5	0.0	6.9	17.6	71.5	0.0	15.3	13.2
		2 mM	75.5	0.0	12	12.5	79.3	0.0	14.9	5.8	76.4	0.0	10.9	12.7
		3 mM	75.4	0.0	11.7	12.9	76.3	0.0	6.0	17.7	70.4	0.0	7.3	22.3
		4 mM	74.6	0.0	6.5	18.9	78.3	0.0	10.9	10.8	76.4	0.0	9.0	14.6
		5 mM	73.9	0.0	10.4	15.7	78.4	0.0	9.5	12.1	73.9	0.0	8.7	17.4
		6 mM	76.5	0.0	14.1	9.4	79.7	0.0	8.7	11.6	73.6	0.0	8.1	18.3
		7 mM	78.6	0.0	8.8	12.6	82.0	0.0	6.9	11.1	76.0	0.0	8.3	15.7
		8 mM	72.7	0.0	6.6	20.7	76.2	0.0	0.5	23.3	72.9	0.0	8.9	18.2
Deoxygenated-Mb		Control (0 mM)	73.3	0.0	12.9	13.8	75.8	0.0	10.5	13.7	70.6	0.0	10.1	19.3
	PYR	1 mM	78.2	0.0	7.7	14.1	80.0	0.0	13.8	6.2	75.1	0.0	7.0	17.9
		2 mM	72.8	0.0	9.7	17.5	70.5	0.0	12.6	16.9	74.2	0.0	8.7	17.1
		3 mM	77.7	0.0	9.9	12.4	73.7	0.0	12.5	13.8	76.4	0.0	10.0	13.6
		4 mM	79.5	0.0	12.4	8.1	78.2	0.0	11.0	10.8	75.2	0.0	6.9	17.9
		5 mM	74.4	0.0	10.5	15.1	77.4	0.0	8.6	14.0	76.7	0.0	8.6	14.7
		6 mM	67.9	0.0	8.4	23.7	72.9	0.0	7.1	20.0	76.5	0.0	12.3	11.2
		7 mM	74.4	0.0	15.7	10.2	77.4	0.0	11.7	10.9	71.3	0.0	10.6	18.1
		8 mM	72.1	0.0	9.2	20.1	73.9	0.0	7.1	19.0	72.6	0.0	5.9	21.5
	LAC	1 mM	82.8	0.0	12.3	4.9	77.7	0.0	9.1	13.2	72.4	0.0	7.9	19.7
		2 mM	68.7	0.0	10.6	20.7	75.7	0.0	7.0	17.3	81.4	0.0	10.0	8.6
		3 mM	78.1	0.0	9.0	12.9	80.7	0.0	12.4	6.9	78.3	0.0	11.1	10.6
		4 mM	73.3	0.0	7.2	19.5	77.3	0.0	8.9	13.8	74.7	0.0	6.1	19.2
		5 mM	73.3	0.0	13.3	13.4	76.3	0.0	5.4	18.3	71.2	0.0	4.8	24.0
		6 mM	70.9	0.0	6.2	22.9	80.9	0.0	10.0	9.1	73.0	0.0	9.9	17.1
		7 mM	78.0	0.0	7.6	14.4	77.7	0.0	11.4	10.9	72.6	0.0	6.0	21.4
		8 mM	69.8	0.0	7.0	23.2	80.2	4.3	9.2	6.3	71.4	0.0	12.6	16.0

Mb: myoglobin; LAC: sodium lactate; PYR: sodium pyruvate.

## Data Availability

Not applicable.

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
