# Peer review of "Myoglobin–Pyruvate Interactions: Binding Thermodynamics, Structure–Function Relationships, and Impact on Oxygen Release Kinetics"

_ijms, 2022, doi:10.3390/ijms23158766_

Round 1
Reviewer 1 Report
The article is good. However, the authors should mention their previous article and the results of the previous article should be compared with the present results.
Thank you very much.
Author Response
Authors thank the reviewer comment. As suggested, we have included the results of our previous article for comparison with the present results in the current manuscript and also cited the reference wherever applicable.
Reviewer 2 Report
Dear Authors,
I reviewed the manuscript (ID: ijms-1841665) entitled: “Myoglobin-Pyruvate Interactions: Thermodynamic Binding, Structure-Function Relationships, and Impact on Oxygen Kinetics”.
The article presents very interesting research results focused on pyruvate interaction with myoglobin and the binding and oxygen release kinetics. In the work presents the results section comprehensively and compare each result with the myoglobin-lactate interaction studies, which was an auto plagiarism problem, but hopefully it has been resolved. A very detailed methodology allows other scientists to repeat or continue their research in this area. Correctly selected methodologies and tools allowed the authors to present the results in an interesting way (especially Figure 4: Docking structures of PYR binding with equine skeletal muscle Mb; Table 2: Changes in secondary structure conformations (%) of Mb with varying concentrations of metabolite, PYR and LAC) and to discuss them.
I recommend this article to be printed in International Journal of Molecular Sciences.
My opinion on the duplication of text in the results section in the submitted paper ““Myoglobin-Pyruvate Interactions: Thermodynamic Binding, Structure-Function Relationships, and Impact on Oxygen Kinetics”, with published paper “Myoglobin Interaction with Lactate Rapidly Releases Oxygen: Studies on Binding Thermodynamics, Spectroscopy, and Oxygen Kinetics” in IJMS 2022, is as follows:
Only the duplications in the Results and Discussion section seem relevant. In other areas, the repetitions concern proper names of institutions, chemical compounds and phenomena, etc., which have no substitutes. The material and methodology are also strictly defined and it is difficult to look for substitutes here - because it is supposed to be a clear description. I understand the authors' intention to broadly and comprehensively describe the results and the discussion on the hypothesis: " How pyruvate and lactate can access myoglobin oxygen stores in the cell during changing conditions”, based on the differences between lactate and pyruvate interaction with oxy- and deoxy-Mb structures.
It is also known that in the discussion we use specific phrases that emphasize the essence of the topic. However, I suggest that the authors review the manuscript again and correct (especially in the Results and Discussion section) those passages that contain large duplicate phrases that are identical to the text in the above-mentioned article.
Author Response
Authors thank the reviewer comment. As correctly mentioned by the Reviewer about the challenges arise in writing specific phrases that might have minor duplication with the article that has similar research design, experimentation and discussion due to the nearly similar metabolites. Although, we have tried to re-write the text without duplication, due to close similarity of the experimental design, approach and outcomes between these two studies made us include few duplication sentences in the text. However, in the revised manuscript, we have removed the duplication sentences in the text to a greater extent in the results and discussion section.
Round 2
Reviewer 1 Report
Thank you very much
Author Response
Point 1.
Paper title should be modified. First, there is no such term "thermodynamic binding" ("equilibrium binding" is often used instead). "Binding thermodynamics" would be more appropriate here. Second, the expression "oxygen kinetics" does not make sense because kinetics should refer to reaction/process, not compound. "Oxygen release/binding kinetics" would be more appropriate. Similar modifications should be done in several places in the body of the manuscript.
Response 1: Authors are thankful for the Editor notes. We agree with the Editor comments and changed the title and also in the body of the manuscript.